# First Report of *Neocucurbitaria unguis*-*hominis* Keratitis

**DOI:** 10.3390/jof9010008

**Published:** 2022-12-21

**Authors:** Nerea Sáenz-Madrazo, Azucena Baeza, Jesús Guinea, Pablo Martín-Rabadán, Alejandro Ruiz-Velasco-Santacruz, José Luis Urcelay

**Affiliations:** 1Department of Ophthalmology, Hospital General Universitario Gregorio Marañón, 28007 Madrid, Spain; 2Department of Immunology, Ophthalmology and Otorhinolaryngology, Faculty of Medicine, Universidad Complutense, 28040 Madrid, Spain; 3Clinical Microbiology and Infectious Diseases Department, Hospital General Universitario Gregorio Marañón, 28007 Madrid, Spain; 4Instituto de Investigación Sanitaria Gregorio Marañón, 28007 Madrid, Spain; 5CIBER Enfermedades Respiratorias-CIBERES (CB06/06/0058), 28029 Madrid, Spain; 6Department of Medicine, Faculty of Medicine, Universidad Complutense de Madrid, 28040 Madrid, Spain

**Keywords:** *Pyrenochaeta*, *Phaeohyphomycosis*, fungal keratitis, coelomycetes, *Neocucurbitaria unguis*-*hominis*, *Pyrenochaeta unguis*-*hominis*

## Abstract

Coelomycetous fungi are among the emerging causes of infections and have been involved in many kinds of infections, including keratitis and endophtalmitis. Here, we present the first case of keratitis caused by *Neocucurbitaria unguis-hominis,* a coelomycetous fungus belonging to the family Cucurbitariaceae. In this case report, we describe the clinical presentation of a 56-year-old woman, a regular contact lens wearer, who was treated for pain in her right eye and fixed spot vision after an injury with plant debris. On examination, a corneal ulcer was observed, the foreign body was removed, and topical eye-drop therapy was started. After an initial improvement, the patient returned three weeks later due to a recurrence of discomfort in her right eye, observing the persistence of the corneal ulcer. Corneal scrapings were taken for culture, growing a filamentous fungus after seven days, which was identified by sequencing the fungal internal transcribed spacer region. It should be noted that microbiological identification of the coelomycetes in the clinical laboratory is not easy because of their difficulty in sporulating, making molecular techniques based on the amplification and sequencing of appropriate phylogenetic markers essential. Identification of these fungi is mandatory in order to optimise treatment due to the difficulty in eradicating them with antifungal treatment, requiring surgery in 50% of cases.

## 1. Introduction

Fungal keratitis is generally exogenously acquired, usually resulting from the traumatic implantation of fungal fragments or spores into the cornea. The most frequent cause is trauma caused by plant debris, which represents the causative mechanism in 60% of fungal keratitis. The increasing use of contact lenses has encouraged the development of fungal keratitis, especially in developed countries [1]. The common cornea pathogenic fungi include species of the genera *Fusarium*, *Aspergillus*, *Candida*, *Curvularia* and *Penicillium*, in terms of frequency depending on geographical factors, occupational and host factors. Nevertheless, a number of fungal species are consistently detected as a cause of keratitis [2,3]. Coelomycetous fungi are among the emerging causes of infections and have been involved in many kinds of infections, ranging from superficial to systemic involvements, including keratitis and endophtalmitis. Traumatic implantation of either contaminated plant debris or conidia-containing soil particles is the most common cause [4].

Here, we present the first case of keratitis caused by *Neocucurbitaria unguis-hominis.* Updating the fungal causes of keratitis in humans is crucial to optimising antifungal treatment for an infection that is complex to treat and often requires therapeutic penetrating keratoplasty.

## 2. Clinical Description

A 56-year-old woman, who is a regular contact lens wearer due to myopia, was treated at the emergency department for pain in her right eye and fixed spot vision after an injury with plant debris (day 0). On examination, she had a visual acuity in her right eye of 20/32 (Snellen). Slit lamp examination revealed peripheral corneal vascularization and an inferior paracentral foreign body resembling plant debris with an infiltration of 1 mm. There was no Tyndall or hypopyon but a mild flare (Figure 1A). The foreign body was removed, and topical eye-drop therapy, including ciprofloxacin and dexamethasone 0.1% (four times daily) and cyclopentolate 1% (topical cycloplegic) (three times daily), was started. Over the next 96 h, the patient’s symptoms significantly improved, but clinical examination revealed a new, small fluorescein-positive area (day four after ocular trauma). The topical cycloplegic was discontinued, and the antibiotics were switched to topical tobramycin (4 times daily) for the next 12 days (day 16).

## 3. Clinical Course and Microbiological Studies

However, on day 23, the patient went back to the emergency department due to a recurrence of discomfort in her right eye. At that time, the examination showed a conjunctiva with mild hyperemia, the cornea being transparent but with the persistence of a fluorescein-positive area, and a lower paracentral corneal ulcer of 1 mm base infiltrated with positive Tyndall and without hypopyon. Corneal scrapings were taken for bacterial, amoebae and fungal cultures; eye-drop treatment was restarted and reinforced with topical tobramycin, ciprofloxacin and voriconazole every 2 h, topical cycloplegic for pain control, and additional systemic doxycycline (100 mg twice a day).

On day 30, a filamentous fungus grew on the cultures reported (Figure 1D). The remaining microbiological cultures resulted in a negative result. At that time, the patient reported symptomatic improvement, but a 1 mm ulcer was detected with fluo+ branching without Tyndall. Pentacam Scheimpflug imaging was also obtained (Figure 1B,C), showing inferior thinning of the ulcer and some calcific precipitates with a 50% infiltration of the corneal thickness. Pending on the definitive fungal species identification, the prior medical treatment was maintained, and the mycotic plaque was removed by surgical scrapings, which caused a residual fungal ulcer.

The fungus was identified as *Neocucurbitaria unguis-hominis* (formerly *Pyrenochaeta unguis-hominis*) by sequencing the fungal internal transcribed spacer region (ITS1-5.8S-ITS2), due to difficulties in achieving an unambiguous identification by morphological examination. The isolate did not grow in RPMI broth medium, and it was not possible to perform antifungal susceptibility testing. Terbinafine, 250 mg every 24 h orally was added to the previous treatment with oral doxycycline and the eye-drop therapy of ciprofloxacin and voriconazole.

Two months after the ocular trauma, the patient remained asymptomatic with a non-infiltrated corneal ulcer of inactive appearance, without hypopyon or Tyndall. Pentacam Scheimpflug imaging showed no significant corneal thinning. Doxycycline was stopped, and topical treatment was spaced out (voriconazole every 4 h and ciprofloxacin every 12 h). One month later (day 90), biochemical analyses showed a gamma-glutamyl transferase (GGT) of 82 IU/L (<40), with the remaining transaminases and bilirubin values being within a normal range, so terbinafine was discontinued two months after its prescription, with a subsequent normalisation of GGT. Four months later (day 210), the patient remained asymptomatic, and the examination showed a residual lesion in the form of corneal leukoma fluo negative with mild corneal thinning with posterior stromal fibrosis but well epithelialised; topical voriconazole was tapered, spacing its administration until its end 8 months after the trauma (day 240).

## 4. Discussion

The coelomycetous fungi constitute a large number of taxa, characterised by the production of conidia (asexual propagules) within a cavity lined by fungal or host tissue, called conidiomata. Coelomycetous fungi are mostly saprobic and parasites of terrestrial vascular plants, but they can also infect vertebrates and other fungi. They are ubiquitous in soil, in salty and freshwater environments, and in sewage [4]. In the south-western European region, the most prevalent species are *Medicopsis romeroi*, *Neocucurbitaria keratinophila*, *Neocucurbitaria unguis-hominis* and *Paraconiothyrium cyclothyrioides* [5].

Pyrenochaeta is a coelomycetous fungus belonging to the family Cucurbitariaceae and the genus Pleosporales. The family Cucurbitariaceae is made up of *Neocucurbitaria cava* (formerly *Pyrenochaeta cava*), *Neocucurbitaria unguis-hominis* (formerly *Pyrenochaeta unguis-hominis*) and *Neocucurbitaria keratinophila* (formerly *Pyrenochaeta keratinophila*) [6]. *Biatriospora mackinnonii* (formerly *Pyrenochaeta mackinnonii*) and *Medicopsis romeroi* (formerly *Pyrenochaeta romeroi*) have recently been reclassified into the families Incertae sedis and Nigrogranaceae, respectively, both belonging to the Pleosporales order [6]. Of these, only *N. keratinophila* [3,7] and *B. mackinnonii* [8] have been implicated in keratitis. The first reference to a case of keratitis caused by *N. keratinophila* was reported in 2009 [7]. Since then, the presence of *P. keratinophila* in the eye has only been identified in a conjunctival exudate [9]. To the best of our knowledge, this is the first report of *Neocucurbitaria unguis-hominis* as a cause of keratitis in a patient and puts that species on the map of fungal causes of keratitis in humans.

Identification of the coelomycetes in the clinical laboratory is not easy because of their difficulty in sporulating [3]. In our case, the isolate did not grow in a specific medium, and we needed sequencing techniques for its identification. Therefore, molecular techniques based on the amplification and sequencing of appropriate phylogenetic markers are critical in the identification of these fungi [6], because they can only be properly identified by DNA sequencing and comparison with reference strains [6].

Microbiological identification of the species is essential for gaining valuable insight about accurate epidemiology and, if possible, for obtaining data on their susceptibility to antifungals [2]. Guidance on antifungal treatments for infections caused by *Neocucurbitaria unguis-hominis* has not been established yet because of the lack of antifungal susceptibility data for these fungi that hardly sporulate [6]. In the present case, as other authors have described, it was not possible to perform antifungal susceptibility testing, as the isolate did not grow in RPMI broth medium. In general, all antifungal agents showed good in vitro activity against the coelomycetous fungi, with terbinafine being the most active [4]. In a study carried out in Spain on eight *N. keratinophila* isolates, all drugs tested showed good in vitro activity, with anidulafungin as the most active agent (MIC_90_ of 0.015 µg/mL), whereas posaconazole was the most active azole (MIC_90_ of 0.125 µg/mL), and isavuconazole being the only drug that showed a poor antifungal activity (MIC_90_ of 2 µg/mL) [9]. A study of eighteen primary cutaneous and/or subcutaneous infections due to coelomycetes showed low MICs of amphotericin B (0.06 to 1 µg/mL), voriconazole (0.03 to 0.5 µg/mL) and terbinafine (0.06 to 1 µg/mL) against all the strains [10]. 

Topical treatment is recommended, and oral systemic treatment may be considered in severe or refractory cases [11]. However, it should be noted that drop-eye therapy is further limited, as many antifungal compounds are fungistatic rather than fungicidal at the concentrations used in topical preparations [12]. For this reason, we support the use of additional systemic antifungal treatments to improve the visual prognosis, as it was the case here reported. Finally, it should be noted that up to 50% of the fungal keratitis cases require therapeutic penetrating keratoplasty (TPK) for the control of infection. In the secondary analysis of the mycotic ulcer treatment trial II which tried to identify patients with infectious keratitis who were at risk of a poor outcome, the presence of hypopyon at baseline was associated with a 2.3-fold increase in the odds ratio of developing corneal perforation and/or needing TPK [13]. Our patient did not present hypopyon, neither at diagnosis nor at follow-up, and, in contrast to previously reported *N. keratinophila* keratitis, no surgical procedure was necessary for the resolution of the fungal infection, demonstrating inactivity of the residual fungal lesion after eight months of follow-up.

## Figures and Tables

**Figure 1 jof-09-00008-f001:**
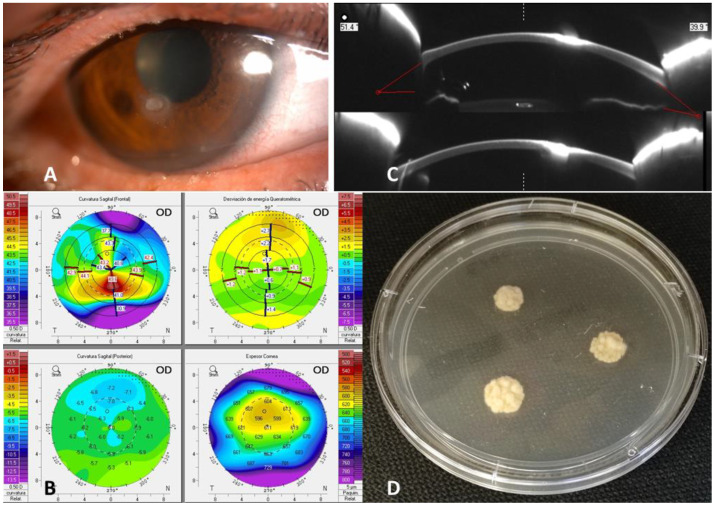
Slit lamp examination showed a dry-looking ulcer with stellate feathery or irregular margins with associated satellite lesions (**A**). Pentacam Scheimpflug imaging was also obtained, showing inferior thinning of the ulcer (**B**) and some calcific precipitates with 50% infiltration of the corneal thickness (**C**). Colonies on Sabouraud-detroxe agar plates (surface) (**D**).

## Data Availability

All data presented in this case report are included in the article.

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
