# Peer review of "First Report of Neocucurbitaria unguis-hominis Keratitis"

_jof, 2022, doi:10.3390/jof9010008_

Round 1
Reviewer 1 Report
In this case report the author describe the first case of keratitis due to Neocucurbitaria unguis-hominis according to the authors. The progression of the infection is easy to read and understand. However, there are certain aspects than need to be solved before being accepted for publication.
Major:
- In my opinion, the discussion completely, except the last 2 paragraphs is a new introduction (with a better quality than the real introduction, I would say). In those paragraphs there is no opposition of your data to the existing literature, nor data interpretation. The whole report is nice, but the discussion is not really such, unfortunately.
Minor:
- I would suggest to mention the genera in line 33 in alphabetical order
- A clear indication of events with their exact day can help to better understand the evolution of the patients. You can set as day 0 either the admission day or the day of diagnosis
Author Response
Reviewer #1 (Comments for Author):
In this case report the author describe the first case of keratitis due to Neocucurbitaria unguis-hominis according to the authors. The progression of the infection is easy to read and understand. However, there are certain aspects than need to be solved before being accepted for publication.
Major:
- In my opinion, the discussion completely, except the last 2 paragraphs is a new introduction (with a better quality than the real introduction, I would say). In those paragraphs there is no opposition of your data to the existing literature, nor data interpretation. The whole report is nice, but the discussion is not really such, unfortunately.
We agree with the reviewer that there was redundancy in the content of the introduction and the first paragraphs of the discussion. We consider that the general description of fungi is more appropriate in the introduction and have therefore modified it. In this way, it would not be necessary to mention the genera in alphabetical order since the order is by frequency, as stated in the text (see minor comment).
We also agree with the reviewer that there were few references to patient's management in the discussion. In this sense, in accordance with the reviewer's suggestion, different sentences have been added to the discussion (underlined throughout the text).
Minor:
- I would suggest to mention the genera in line 33 in alphabetical order
As described above, this paragraph has been deleted.
- A clear indication of events with their exact day can help to better understand the evolution of the patients. You can set as day 0 either the admission day or the day of diagnosis
In the previous version we tried to provide a chronological order to facilitate the reading of the case report. However, we agree with the reviewer that using the nomenclature of day 0 as the first day of attendance makes it easier for readers to follow the evolution. Therefore, we have added that suggestion.
Reviewer 2 Report
Dear Authors,
I am glad to review the paper titled: "First report of Neocucurbitaria unguis-hominis keratitis. "
The paper is well-written, and the Authors should be commended for their work. The case is well-reported, and the discussion highlights what was previously explained regarding the case report's findings.
Nonetheless, summarizing "the case report presentation" in the abstract would be helpful for readers.
Author Response
Reviewer #2 (Comments for Author):
Dear Authors,
I am glad to review the paper titled: "First report of Neocucurbitaria unguis-hominis keratitis. "
The paper is well-written, and the Authors should be commended for their work. The case is well-reported, and the discussion highlights what was previously explained regarding the case report's findings.
Nonetheless, summarizing "the case report presentation" in the abstract would be helpful for readers.
First of all, thank you for your kind words. The reviewer considers appropriate to summarise the case report in the abstract to make it easier to read. We agree that it is useful information for the reader, so we have modified the abstract, reducing some of the previous information to fit into the space available. Please find attached the new abstract for your review:
“Coelomycetous fungi are among emerging cause of infections and have been involved in many kinds of infections, including keratitis and endophtalmitis. Here we present the first case of keratitis caused by Neocucurbitaria unguis-hominis, a coelomycetous fungi belonging to the family Cucurbitariaceae. In this case report we describe the clinical presentation of a 56-year-old woman, regular contact lens wearer,was cared for pain in the right eye and fixed spot vision after get injured with plant debrids. On examination, a cornel ulcer was observed, the foreign body was removed and topical eye-drop therapy was started. After an initial improvement, the patient returned 3 weeks later due to recurrence of discomfort in her right eye, observing the persistence of the corneal ulcer. Corneal scraping were taken for culture, growing a filamentous fungus after 7 days, which was identified by sequencing the fungal internal transcribed spacer region. It should be noted that microbiological identification of the coelomycetes in the clinical laboratory is not easy because of their difficulty in sporulating, making molecular techniques based on the amplification and se-quencing of appropriate phylogenetic markers essential. Identification of these fungi is mandatory to optimise treatment due to the difficulty in eradicating them with antifungal treatment, requiring surgery in 50% of cases.”
Round 2
Reviewer 1 Report
no further comments. I suggest to be accepted